Corrected: Author correction

# Sculpting nanoparticle dynamics for single-bacteria-level screening and direct binding-efficiency measurement

Y.Z. Shi[1,2], S. Xiong[2], Y. Zhang [3], L.K. Chin[2], Y.-Y. Chen[2], J.B. Zhang[2], T.H. Zhang[4], W. Ser[2],
A. Larrson[5], S.H. Lim[5], J.H. Wu[1], T.N. Chen[1], Z.C. Yang[6], Y.L. Hao[6], B. Liedberg[7], P.H. Yap[8], K. Wang[9,10],
D.P. Tsai[11], C.-W. Qiu[4,12] & A.Q. Liu[2,6]

Particle trapping and binding in optical potential wells provide a versatile platform for various biomedical applications. However, implementation systems to study multi-particle contact interactions in an optical lattice remain rare. By configuring an optofluidic lattice, we demonstrate the precise control of particle interactions and functions such as controlling aggregation and multi-hopping. The mean residence time of a single particle is found considerably reduced from 7 s, as predicted by Kramer's theory, to 0.6 s, owing to the mechanical interactions among aggregated particles. The optofluidic lattice also enables single-bacteria-level screening of biological binding agents such as antibodies through particle-enabled bacteria hopping. The binding efficiency of antibodies could be determined directly, selectively, quantitatively and efficiently. This work enriches the fundamental mechanisms of particle kinetics and offers new possibilities for probing and utilising unprecedented biomolecule interactions at single-bacteria level.

[1] School of Mechanical Engineering, Xi'an Jiaotong University, Xi'an 710049, China. [2] School of Electrical and Electronic Engineering, Nanyang Technological University, Singapore 639798, Singapore. [3] School of Mechanical and Aerospace Engineering, Nanyang Technological University, Singapore 639798, Singapore. [4] Department of Electrical and Computer Engineering, National University of Singapore, Singapore 117583, Singapore. [5] School of Biological Sciences, Nanyang Technological University, Singapore 639798, Singapore. [6] National Key Laboratory of Science and Technology on Micro/Nano Fabrication, Institute of Microelectronics, Peking University, Beijing 100871, China. [7] Centre for Biomimetic Sensor Science, School of Materials Science and Engineering, Nanyang Technological University, Singapore 639798, Singapore. [8] Lee Kong Chian School of Medicine, Nanyang Technological University, Singapore 308232, Singapore. [9] College of Biomedical Engineering, Taipei Medical University, Taipei 11031, Taiwan. [10] Nanyang Environment & Water Research Institute, Nanyang Technological University, Singapore 637141, Singapore. [11] Department of Physics, National Taiwan University, Taipei 10617, Taiwan. [12] SZU-NUS Collaborative Innovation Center for Optoelectronic Science and Technology, Shenzhen University, Shenzhen 518060, China. Correspondence and requests for materials should be addressed to S.X. (email: xiongsha@ntu.edu.sg) or to C.-W.Q. (email: eleqc@nus.edu.sg) or to A.Q.L. (email: eaqliu@ntu.edu.sg)

Particle hopping between optical potential wells has attracted attention owing to its extensive involvement in many physical and biological processes, such as cell and DNA stretching[1–3], protein folding[4–7], chemical reactions[8,9] and biomolecule sorting[10–13]. Following the pioneering work of Kramer in the 1940s, the random motion and escape rate of particles from a potential well have been studied extensively[14–16]. Thermally activated particle hopping between neighbouring potential wells is reported in the archetypal dual optical traps with symmetric[17–20] or asymmetric[21] potential distributions or in dual nanohole traps[22], and complies closely with Kramer's theory. Particle hopping is also investigated in optical line traps formed by two counter-propagating waves[23] or an array of optical traps[24]. Nano-optical conveyor belts that transport nanoparticles to a desired position by particle hopping have been experimentally demonstrated by varying the potential wells through tuning the wavelength or polarisation of light[25–27] or applying external stimuli[28,29]. Particle hopping has also been investigated in two-dimensional (2D)[30–32] and three-dimensional (3D)[33] potential landscapes. However, majority of works hitherto focus on the hopping of an individual particle between the potential wells, whereas the rich degrees of freedom in particle–particle interactions have been neglected. Although non-contact interactions such as optical binding[34,35] have been investigated, contact interactions, such as particle collision and particle aggregation in the optical lattice, are considered futile due to the lack of clear-cut paradigms to meaningful applications. Nevertheless, such perception is not true. The rich physics behind the multiple particle contact interactions still remains an enigma.

Apart from particle hopping, optical potential wells are promising for single-cell trapping to screen biological binding agents such as antibodies[36,37], peptide or aptamers[38,39], which play a crucial role in pathogen recognition and inhibition in diagnostics and therapeutics of infectious disease[40,41]. Single bacteria isolation and detection is an emerging technique because the heterogeneity between individual bacteria cannot be revealed by conventional bulk approaches[42]. Therefore, it is highly desired to screen the biological binding agents, such as antibodies, at single-cell level to reveal new insights of complex biological interactions. However, such needs are not fulfilled by conventional binding assays such as a precipitation assay[43], agglutination assay[44], enzyme-linked immunosorbent assay (ELISA)[45,46], surface plasma resonance (SPR)[47,48], western blot[49] and fluorescence-activated cell sorting (FACS)[50]. ELISA generates colorimetric or fluorescent signals and evaluates the quantity of sample input by interpolating against a standard curve. SPR determines the dissociation constant ($K_d$), which is a measurement of the binding affinity. However, these two assays require a relatively large number of bacterial cells and delicate processing of multiple step reactions. The bacteria loads in clinical samples are often too low for direct detection. A lengthy bacterial culture would be required to enrich the sample for detection, which takes half a day to one day. FACS is a technique capable of counting the binding efficiency between stained bacterial cells and microparticles. However, the extra staining process may interfere with the downstream assays and the microparticles are likely to affect the signal readout. Furthermore, multiple cells probably bind to the same microparticle and be counted as one.

Here, we establish a holistic framework for steering the kinetics and synergies between multiple particles, enabled by the tunable control of the optical and hydrodynamic forces in an optofluidics lattice. Particles are driven between hotspots by both optical and hydrodynamic forces through different hopping mechanisms, i.e., particle bypassing, collision and aggregation. Thanks to the particle–particle interaction, the particle residence time in the potential well is more than one order of magnitude shorter than that predicted by classical Kramer's theory. By delicately controlling optical and hydrodynamic forces, a two dimensional closed-loop trajectory for particle hopping is created in the microchannel, representing a new technique for harnessing Brownian force to create microscopic motors, which has been realised using other forces, such as heating-induced fluidic force[51], photophoretic force[52], optical force[51,53], acoustic force[54], etc. Besides, the optical potential wells enable the trapping of individual bacterial cells in the microfluidic channel to screen biological binding agents, and evaluate the binding affinity and specificity at single bacteria level. With our single-cell approach, we aim to work directly with clinical samples, which have low bacterial load, and shorten the turnaround time for potential diagnostic applications.

## Results

**Hopping mechanisms in the optofluidic lattice**. In contrast to the conventional optical lattice, our optofluidic lattice is classified as a type of discrete optical interference pattern[55] in a microfluidic flow stream. In the optofluidic chip as shown in Fig. 1a, particles are hydrodynamically focused by three flow streams and conveyed by optical and fluidic forces in a discrete optical interference pattern, serving an ideal paradigm for the study of particle interactions in optical landscapes[56–58]. The details of the optofluidic chip are discussed in the Methods section. Particles are injected from the central flow stream and confined by the sheath flow. The discrete optical interference pattern forms an array of hotspots (or potential wells). When a particle is trapped in a hotspot (Fig. 1b), the optical scattering force $F_{scat}$ acts on the particle along the light-propagation direction, counteracting with the drag force along the flow direction. Particles and bacteria can be trapped in the microchannel based on the same principle as the optical chromatography[59–61]. The optical gradient force $F_{grad}$ drives the particle to the position with the highest light intensity, causing a stable trapping. Meanwhile, the Brownian force $F_{brow}$ causes random movement of the particle in all directions, triggering most of the hopping. The particle movement, including hopping, can be steered in the optofluidic lattice containing four hotspots as shown in Fig. 1c. By delicately balancing the preceding forces, particles would hop between the four hotspots, and even form a closed loop around the illustrated region. Detailed information on particle movement in a loop can be found in Supplementary Figs. 1–3.

Three hopping mechanisms are unveiled in the optofluidic lattice: particle bypassing (Fig. 1d), snookering-like collision (Fig. 1e) and aggregation (Fig. 1f). To study the hopping mechanisms, the stable trapping point of the particle on hotspot 1 is made close to hotspot 4 by adjusting the preceding forces. A potential well is created near the edge of hotspot 1 at $P_1$ ($x_1, y_1, z_1$) with the injection point of the laser in the microchannel being set as the zero point $P_0$ (0, 0, 0) as shown in Fig. 1a. Since the hopping between two adjacent hotspots happens along the $x$ direction, the potential energy profile is plotted along the $x$-axis at $y = y_1$ and $z = z_1$ in the bottom row of Fig. 1d–f. Once a particle (marked in red) is trapped in a potential well, an extra trapping position is induced due to the focusing of light by the trapped particle as shown in Fig. 1d. A second particle (marked in green) bypasses the first one without making contact, and becomes trapped in the induced optical trap. Once this particle is trapped in the induced trap, it easily hops to the adjacent potential well because the original well is too shallow to trap the particle for a long period (bottom of Fig. 1d). This hopping induced by particle bypassing occurs only when the green particle is initially positioned on the top edge of the hotspot ($|y| > 0$). Particle collision occurs when the second particle (green) is on the same

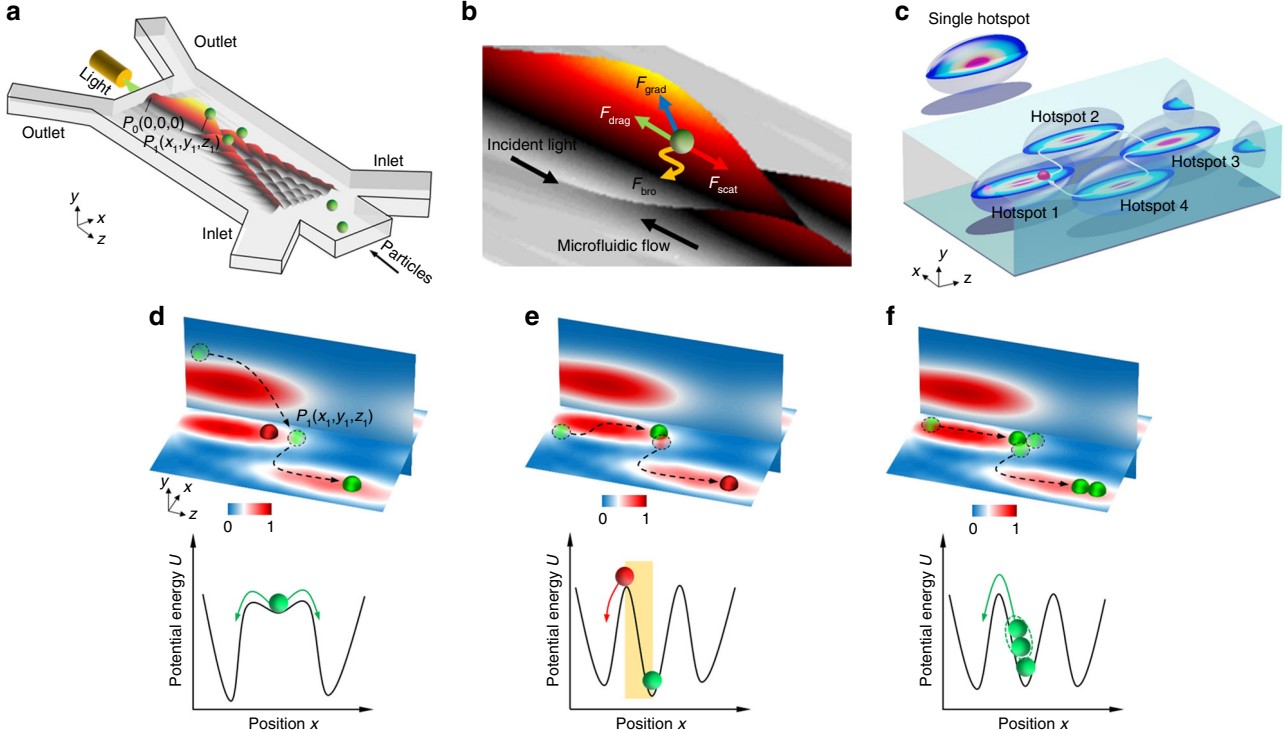

**Fig. 1** The 2D controllable particle hopping in an optofluidic lattice. **a** Generation of the 2D lattice in an optofluidic chip. **b** Forces acting on a particle in the hotspot. **c** Illustration of the realisation of controllable particle hopping loop around the ellipsoid hotspots. Illustration of the simulated normalised light intensity (top row) and potential energy (bottom row) profiles of particle hopping triggered by three different mechanisms: **d** particle bypassing, **e** snookering-like particle collision and **f** particle aggregation. The intensity profiles in $y$–$z$ plane (vertical) are slightly shifted from the central line ($x = 0$) to have a better view of the particle trajectories

$x$–$z$ plane as the pre-trapped particle (red) as shown in Fig. 1e. The green particle is gradually attracted to the central line of the potential well, and collides with the red particle, pushing it to the edge. The collision eventually forces the red particle to hop to the adjacent potential well. During the collision, the red particle is pushed to the saddle of the potential well and falls into the adjacent well easily as shown in the bottom of Fig. 1e. When more particles attract towards the potential well and make a head-on contact with the pre-trapped particle, they aggregate and gradually build up instability in the potential well. Most of the aggregated particles are not trapped at the bottom of the potential well because of the physical contact, and they will hop over the barrier of the potential well after a short residence time. Eventually, only one particle is left in hotspot 1 and trapped stably as shown in Fig. 1f.

**Particle bypassing-induced hopping**. To illustrate the particle–particle interaction in the optical potential well, a 1 μm polystyrene particle (red) is pre-trapped in the first hotspot as shown in Fig. 2a. A second particle (green) is initially placed on the left side of this hotspot and driven by both the fluidic drag force and the optical scattering force towards the red particle. The green particle enters the trap without contacting the red particle as it travels along the edge of the hotspot ($|y| > 0$) as shown in Fig. 2b. Note that the hotspot has a much stronger gradient in the $x$-direction than in the $y$-direction (see Supplementary Figs. 4 and 5 for the details of the lattice), and the trajectories of the green particle with different initial positions are depicted in Supplementary Fig. 6. The extra trapping position for the green particle, which is 3.5 μm from the trapping position of red particle ($z = 23.5$ μm), is induced by the light focused by the red particle

(Fig. 2c). It has a shallow potential well with an energy barrier of $2.25 \times 10^{-20}$ J ($\sim 5\,k_B T$), in which the thermal fluctuation of the green particle can easily exceed the depth of this well, enabling the particle to hop to the adjacent potential well in the $x$-direction[62–64].

The mean first-passage time (MFPT) characterises the mean time taken by a particle to jump over the barrier of the potential well, also known as the residence time of the particle in that well. Detailed calculation of MFPT of the green particle in the extra trapping position is discussed in the Methods section and Table 1. The experimentally measured residence time is approximately 7 s, which agrees with the calculated value (8 s). The experimental residence time is obtained based on 20 experimental events, in which the time ranges from 5 to 9 s. The particle trajectory is plotted in Fig. 2d, which shows that the green particle takes 2 s to move to the induced trapping position and stays there for approximately 5 s before hopping to the adjacent potential well. More experimental results unveil that the resident time could reach 9 s. The vibration of the trajectory in the $x$-direction is due to random Brownian motion. Movie illustration of the experimental observation of the 1 μm polystyrene particle hopping from the induced trapping position is shown in Fig. 2e (see also Supplementary Movie 1). The 1 μm particle pre-trapped in the potential well is shown in red. The second particle (green) is pushed towards the red particle by the optical scattering force ($\sim 2$ pN). The green particle passes directly over the red particle, and is trapped in the position induced by the pre-trapped particle for 5.1 s before hopping to the adjacent top potential well. Throughout the process, the red particle remains undisturbed and stably trapped at its original position because of the high barrier of the potential well. The laser power used is 300 mW and the flow velocity is 100 μm s$^{-1}$.

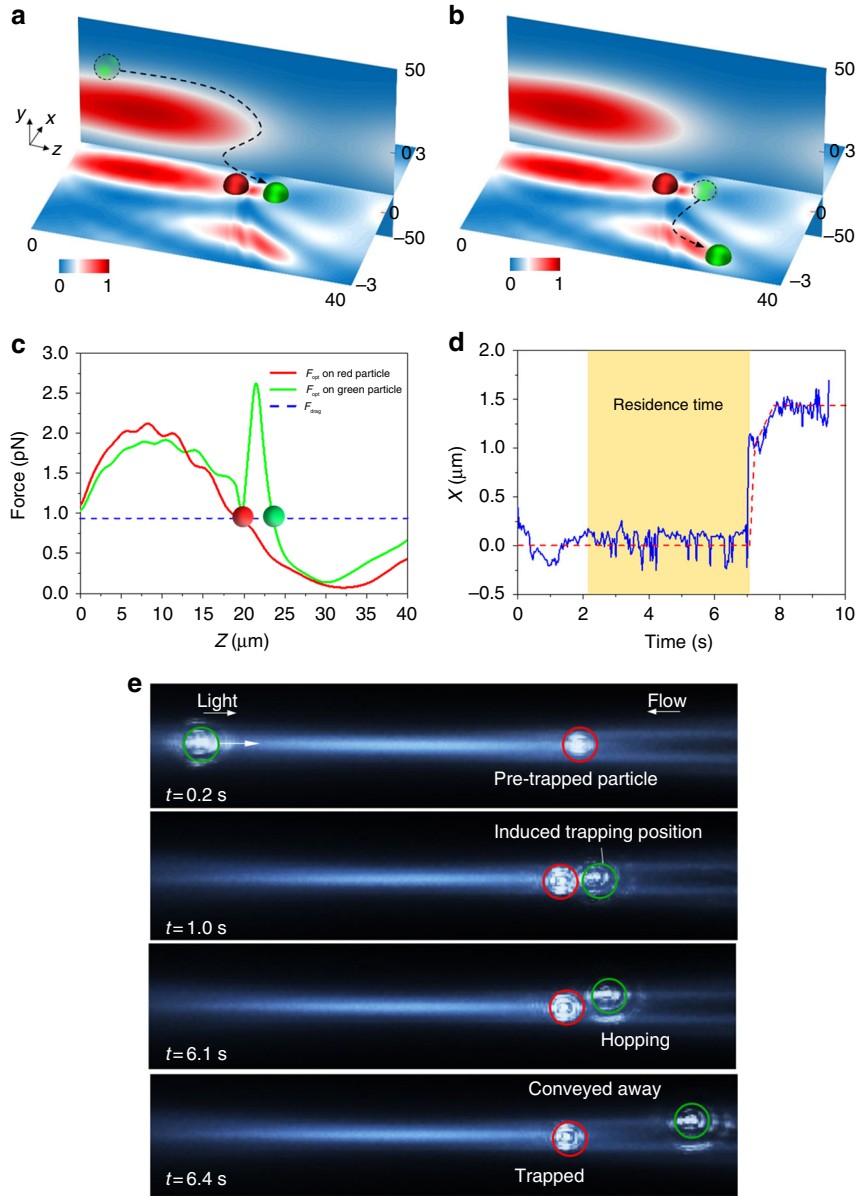

**Fig. 2** Particle hopping from the extra trapping position caused by particle pre-trapping. Illustration of **a** the green particle being trapped in the extra trapping position and **b** hopping to the adjacent potential well. Optical intensity profiles at the $x$–$z$ plane ($y = 0$) and $y$–$z$ plane ($x = 0$) are plotted. **c** Force analysis on the 1 μm polystyrene particle along the $z$-direction when $x = 0$, $y = 0$. Blue line represents the drag force. Red and green lines represent the optical forces on red and green particles, respectively. **d** Projection of the green particle trajectory on the $x$-axis. **e** Experimental demonstration of the 1 μm particle hopping induced by the pre-trapped particle

**Snookering-like collision-induced hopping**. For the case of particle collision-induced hopping, when the second (green) particle originates from the $x$–$z$ plane ($y = 0$), it tends to be swiftly confined in the centre of the beam ($x = 0$) by the optical gradient force as shown in Fig. 3a and Supplementary Fig. 6. The green particle is pushed to the right by the optical scattering force in the $z$-direction. Simultaneously, thermal fluctuation causes the green particle to vibrate in the $x$-direction. The green particle eventually collides with the pre-trapped red particle. This side-on collision can drive the red particle into the adjacent potential well (Fig. 3b). Before the collision, the red particle is trapped at the valley point of the potential well's centre at point $\alpha$ (Fig. 3c), i.e., at $x = 0$, $y = 0$ and $z = 20$ μm. The activation energy of the potential barrier between points $\alpha$ and $\beta$ is $7.5 \times 10^{-19}$ J, which is 181 times the thermal energy of the red particle ($k_B T$). Therefore, the red particle is stably trapped in the potential well. Hopping

occurs only when the red particle crosses the saddle point $\beta$ of the potential well. When the green particle collides with the red particle, the distance between the two particles is almost the same as that between the bottom and saddle points of the potential well (Supplementary Figs. 7 and 8). Once the green particle occupies the valley point (point $\alpha'$) of the new potential well, the red

**Table 1 Parameters for calculating the residence time of the particle in the potential well at the extra trapping position**

| $U$ (A) (J) | $U$ (B) (J) | $U''$ (A) | $U''$ (B) | Calculated time (s) | Measured time (s) |
|---|---|---|---|---|---|
| $5.23 \times 10^{-20}$ | $7.48 \times 10^{-20}$ | $5.78 \times 10^{-7}$ | $-2.19 \times 10^{-6}$ | 8 | $7 \pm 2$ |

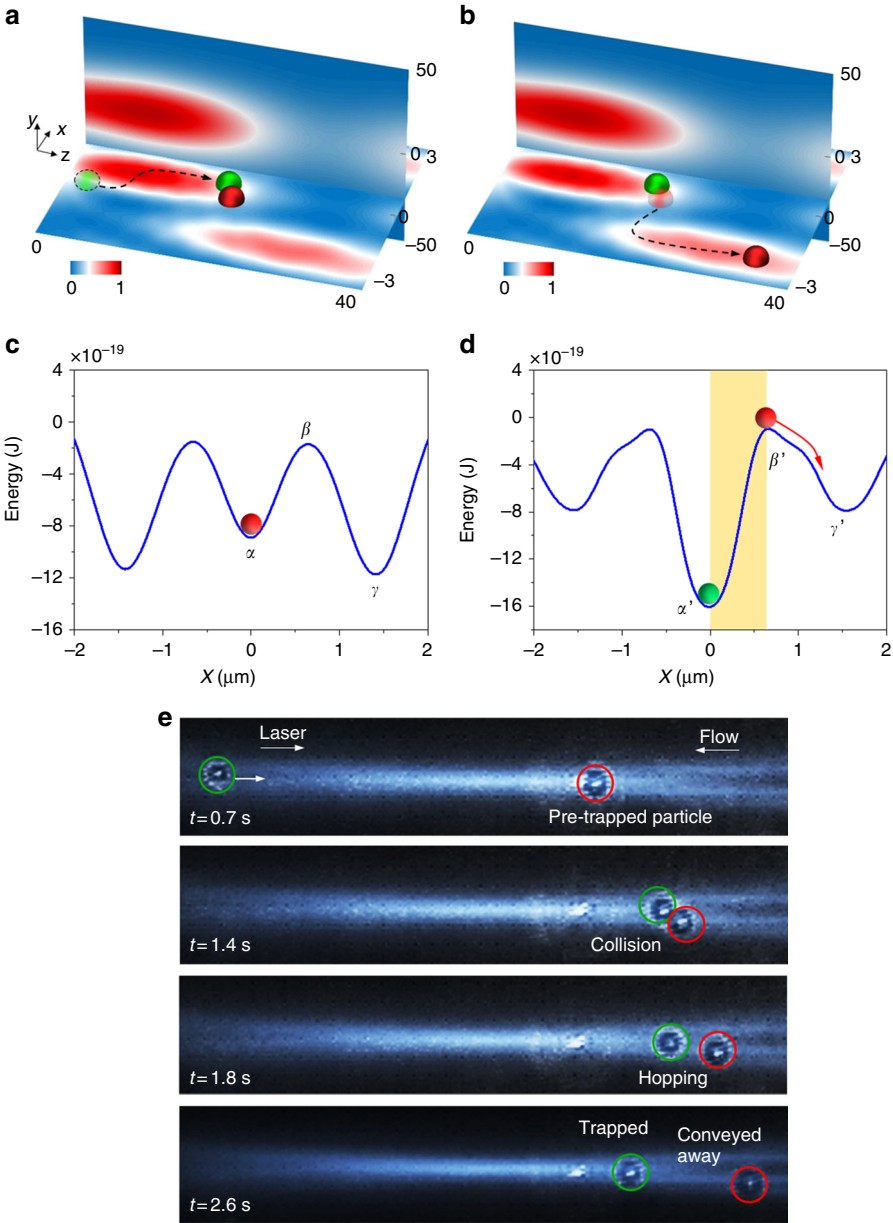

**Fig. 3** Snookering-like particle collision-induced hopping between potential wells. Illustration of the particle **a** collision and **b** hopping combining with optical intensity profiles of the lattice. The energy profile of the potential wells **c** before and **d** after the particle collision, plotted along x-axis at $y = 0$ and $z = 20$ μm. **e** Experimental demonstration of the particle collision-induced hopping between potential wells

particle is pushed over the barrier of the adjacent potential well (point $\beta'$) as shown in Fig. 3d. Consequently, the red particle is easily drawn down into the adjacent potential well with a stable trapping position at point $\gamma'$. The experimental demonstration of this collision-induced hopping in the optofluidic lattice is shown in Fig. 3e (Supplementary Movie 2). Interestingly, when the green and red particles make contact at $t = 1.4$ s, they are shifted to the right-hand side of the original trapping position of the red particle. This movement possibly results from the inertia of the green particle and the greater optical forces acting on them as a large combined object because the optical scattering force increases exponentially with particle size. After hopping at $t = 1.8$ s, they are separately trapped in two different potential wells. The green particle is shifted some distance to the left because of the smaller optical scattering force on a single particle. Different from the case in Fig. 2e, the collision in Fig. 3e induces immediate hopping

of the pre-trapped red particle, after which the green particle replaces the red one in the same trapping position.

**Particle aggregation-induced hopping**. Occasionally, when two particles are both near the centre of the optical potential ($x = 0$), they make contact and tend to aggregate. Aggregation considerably reduces the residence time of the particle located off the centre of the potential well, termed the 'outer' particle. In contrast, the particle near the centre of the potential well is termed the 'inner' particle. Figure 4a shows that the optical and molecule binding forces cause the red and green particles to move together as an aggregated particle in the x-direction. In the potential well, the green and red particles experience optical forces $F_1$ and $F_2$, respectively. Therefore, the total force on the centre of mass of the aggregated particle is expressed as $F_t = F_1 + F_2$. The mean

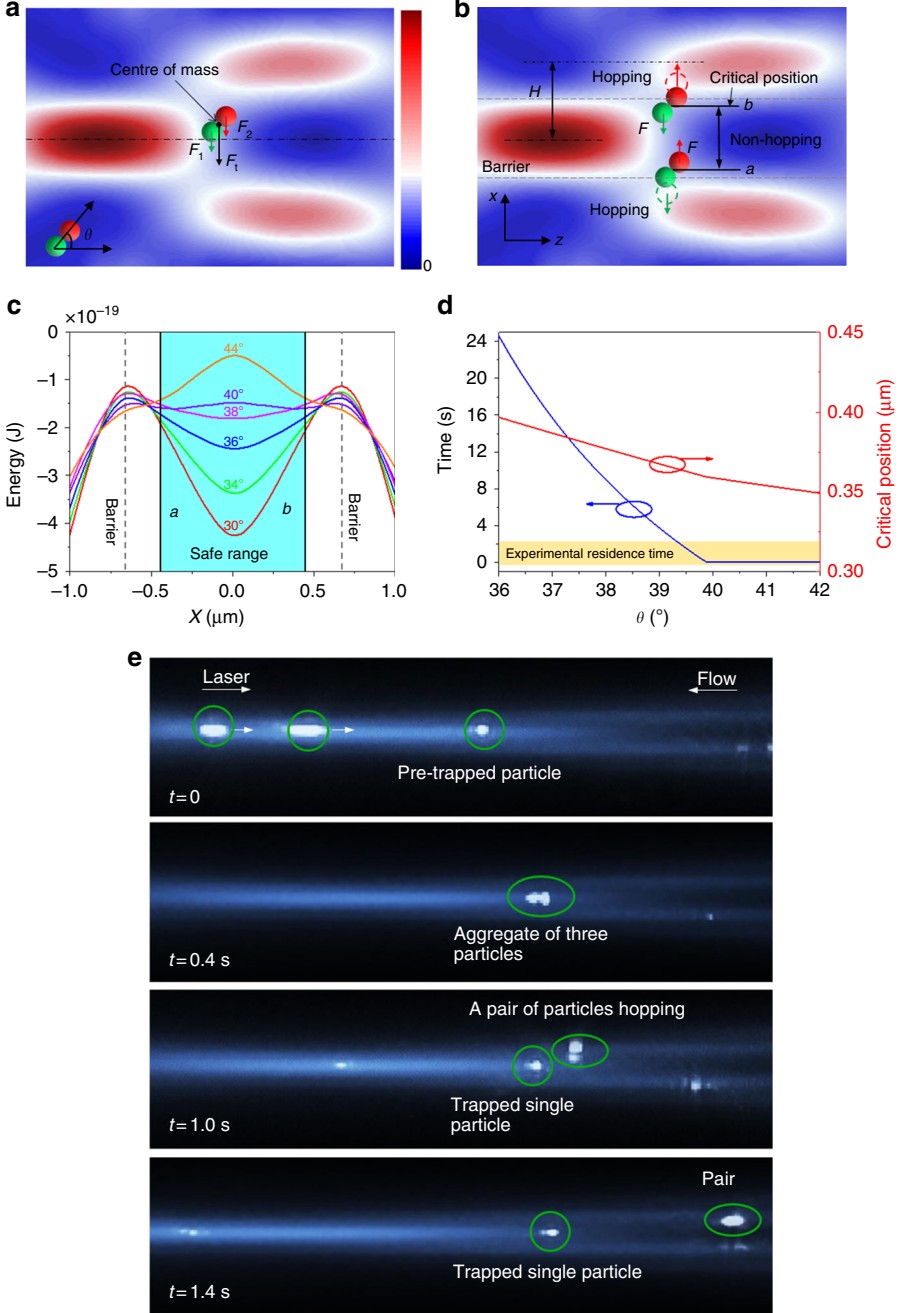

**Fig. 4** Particle hopping induced by particle aggregation. **a** Model for calculating mean first passage time of the particle in the potential well. **b** Illustration of hopping of the particle pair between adjacent hotspots. **c** Potential wells of total force $F_t$ plotted along x-axis at $y = 0$ and $z = 20\,\mu m$ with different contact angles. **d** Calculated mean first passage time of the red particle varies with contact angle θ. Experimental residence time is distributed in the yellow area. **e** Experimental demonstration of the particle hopping induced by particle aggregation in the optofluidic lattice

residence time of the aggregated particle can be regarded as the mean time of the aggregated particle moving from the potential bottom ($x = 0$) to the critical positions $a$ and $b$, whereby the outer particles (red in the top conjunction and green in the bottom conjunction) reach the saddle of the potential wells as shown in Fig. 4b, c. Therefore, the safe range for the trapping of aggregated particle in the potential well is constrained within $a$ and $b$. The contact angle ($\theta$) of the two particles could dramatically affect the profile of potential energy and critical positions, which further change the mean residence time of aggregated particle. The critical position $b$ can be expressed as $b = H/2 - 0.5R\sin\theta$, where $H$ is the distance between the centres of two adjacent hotspots in x-

direction and $R$ is the radius of red particle. Similarly, the critical position $a$ can be expressed as $a = -b = -H/2 + 0.5R\sin\theta$. In our experiments, $H$ is 1.4 μm. The profiles of the potential wells located at particle pre-trapping position ($y = 0$, $z = 20\,\mu m$) with different contact angles are plotted in Fig. 4c. When $\theta < 40°$, there is a valley point at $x = 0$ in each potential well, meaning the aggregated particle can be trapped around the centre of the potential well in the range of $a$ to $b$ for a certain of time. The valley point at $x = 0$ becomes a saddle point when $\theta \geq 40°$, which means that the aggregated particle becomes unstable in the potential well, and the outer particle will immediately hop to the adjacent potential well. The relation of critical point $b$ and contact

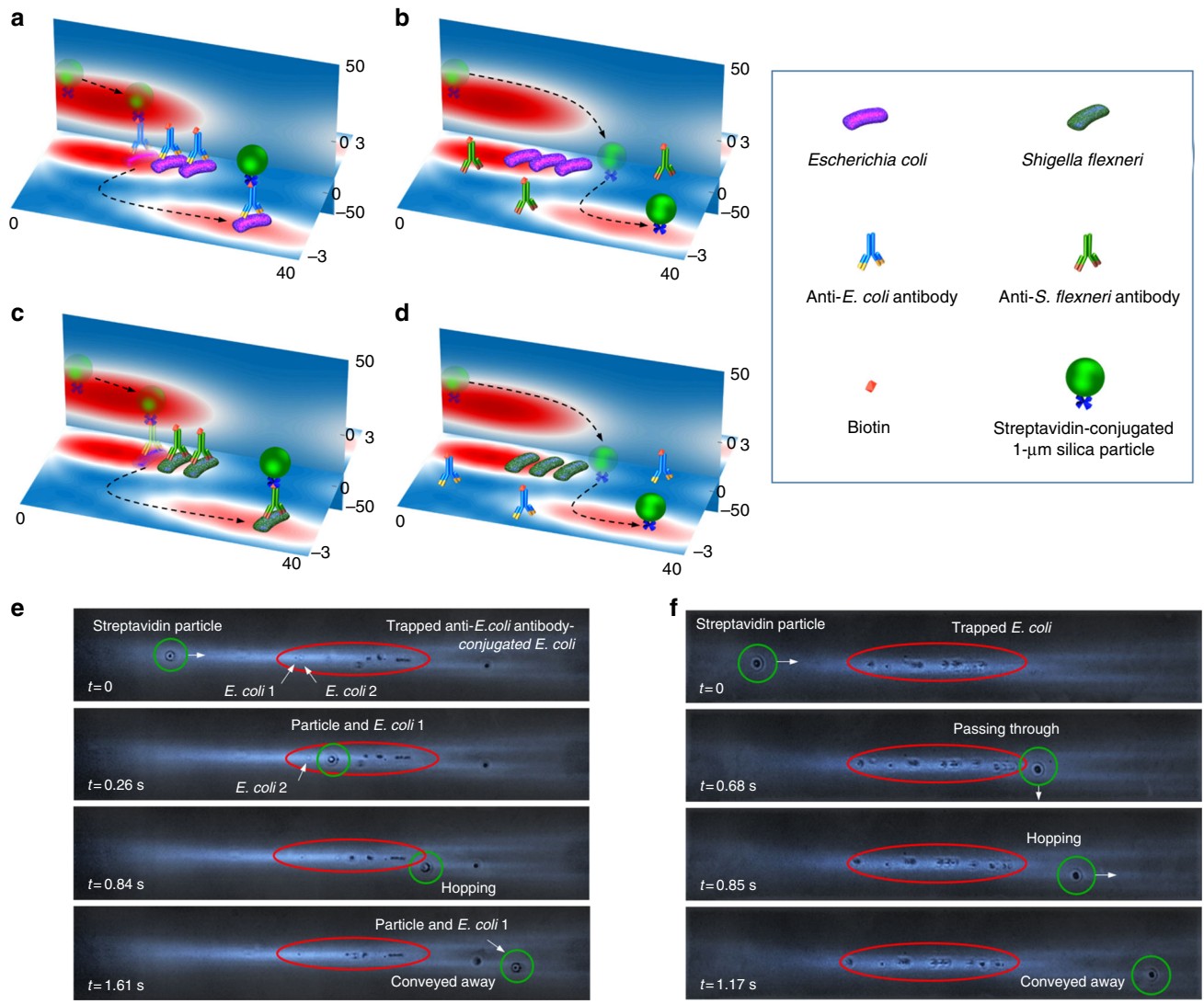

**Fig. 5** Antibody selection through specific binding and hopping. *E. coli* and *S. flexneri* are trapped in the hotspot after they are conjugated with biotin-labelled anti-*E. coli* antibody and biotin-labelled anti-*S. flexneri* antibody, respectively. When a streptavidin-conjugated microparticle passes through the trapped **a** *E. coli* and **c** *S. flexneri* stained with specific antibodies, the microparticle is anchored to the bacterium through the very strong biotin–streptavidin interaction. The microparticle and bacterium hop together to another hotspot. When **b** *E. coli* and **d** *S. flexneri* are trapped in the hotspot after they are incubated with non-specific antibodies, the streptavidin-conjugated microparticle hops to another hotspot without binding with any bacterium due to the unsuccessful labelling of biotin. Experimental observation of the streptavidin-coated microparticle hopped with **e** *E. coli* conjugated with anti-*E. coli* antibody, and **f** passed away from bare *E. coli* due to unsuccessful conjugation of non-specific antibody (anti-*S. flexneri* antibody)

angle is plotted in Fig. 4d. The time needed for the aggregated particle to reach the critical point $a$ or $b$ from $x = 0$ is expressed as

$$t = \frac{\zeta}{k_B T \int_a^b \frac{dx}{\psi(x)}} \left[ \left( \int_a^0 \frac{dx}{d\psi(x)} \right) \int_0^b \frac{dx}{\psi(x)} \int_a^x dx' \psi(x') - \left( \int_0^b \frac{dx}{d\psi(x)} \right) \int_a^0 \frac{dx}{\psi(x)} \int_a^x dx' \psi(x') \right], \quad (1)$$

where $\zeta$ is the fluidic damping constant with $\zeta = 6\pi\eta R'$, where $\eta$ is the dynamic viscosity of the liquid and $R'$ is the hydrodynamic radius of aggregated particle[65]. $k_B$ is the Boltzmann constant and $T$ is the temperature. $\Psi(x)$ is the spatial probability density, expressed as $\psi(x) = \exp[-U(x)/(k_B T)]$, where $U(x)$ is the potential energy as a function of the position of aggregated

particle, which can be expressed as

$$U(x) = \int \left[ F_1 \left( x - \frac{1}{2} r \sin \theta \right) + F_2 \left( x + \frac{1}{2} r \sin \theta \right) \right] dx. \quad (2)$$

The residence time of aggregated particle decreases exponentially with contact angle as shown in Fig. 4d.

Solving the equation for the residence time shows that hopping is predicted to occur instantaneously when the contact angle reaches 40°. In the experiment, the observed residence times are ≤2 s (yellow area of Fig. 4d), which imply that hopping occurs when the contact angle is equal or larger than 39°. The process of particle aggregation-induced hopping is presented in Fig. 4e and Supplementary Movie 3. Initially, two particles travel towards the pre-trapped particle along the central line of the hotspot. Upon contact, all three particles become trapped and aggregated. After 0.6 s, two particles hop together as a pair into the adjacent potential well, leaving a single particle in the original trapping

**Table 2 Binding efficiency for bacteria with different antibodies (Sample size: 300 events)**

| Condition | Bacteria hopping (%) | Bacteria trapping (%) |
|---|---|---|
| *E. coli* | 0 | 100 |
| *E. coli*+anti-*E. coli* antibody | 93 | 7 |
| *E. coli*+anti-*S. flexneri* antibody | 3 | 97 |
| *S. flexneri*+anti-*S. flexneri* antibody | 87 | 13 |
| *S. flexneri*+anti-*E. coli* antibody | 0 | 100 |

position. The residence time of the aggregated particles is much shorter than that of a single particle. The possibility of aggregation-induced hopping implies that when the size of particle is very close to the width of potential well, each well in the optical lattice can only stably trap one particle at a time; extra particles tend to aggregate and hop away because of the physical contact of particles in fluid.

**Antibody screening and binding efficiency measurement**. We propose to screen antibodies of bacteria at single bacteria level for their binding affinity and specificity using our platform. Individual bacterial cells were first conjugated with biotin-labelled antibodies and, subsequently, trapped in the optical potential wells. Then, a streptavidin-conjugated microparticle passed by and bound to the bacteria. The microparticle–bacterium complex would hop away from the potential well. With high-affinity antibodies, a large percentage of bacterial cells would bind to the microparticles and hop away and vice versa. For antibodies with high specificity (Supplementary Figs. 9 and 10), only the target bacteria would hop away with the microparticle whereas non-target bacteria remained trapped. In contrast, antibodies with low specificity would bind to non-target cells, causing the non-target cells to hop away from the potential well.

As a proof of concept, we evaluated the binding affinity and specificity of two antibodies, one against *Escherichia coli* (*E. coli*) and one against *Shigella flexneri* (*S. flexneri*). The working principle of the hopping mechanism in the optofluidic chip is shown in Fig. 5a–d. The bacterial cells were trapped in the first hotspot when the laser power was 400 mW and the flow velocity was 50 µm s$^{-1}$. The magnitude of the maximum optical forces on the 1 µm polystyrene particle and *E. coli/S. flexneri* are approximately 20 and 1 pN, respectively. The drag force, which is proportional to the diameter of particle, is approximately 0.5 pN. Therefore, *E. coli* can be trapped on hotspot 1 while the optical scattering force pushes the particle to the edge of the hotspot. When *E.coli* were conjugated with biotin-labelled anti-*E. coli* antibodies, the 1 µm silica microparticle captured a single bacterial cell (Fig. 5a), and the microparticle–bacterium complex hopped together to the adjacent hotspot where the potential well was deeper and stronger. When *E. coli* were mixed with the non-specific antibodies (anti-*S. flexneri*), the streptavidin-conjugated microparticle bypassed the bacteria and hopped to the adjacent hotspot on its own as shown in Fig. 5b. The selective binding was also observed with *S. flexneri*. Similar to Fig. 5a, b, the microparticle bound to and hopped with *S. flexneri* cell conjugated with anti-*S. flexneri* antibody, but hopped on its own when *S. flexneri* was mixed with the non-specific (anti-*E. coli*) antibody. Experimental demonstrations of the selective binding and hopping of *E. coli* incubated with specific and nonspecific antibodies were shown in Fig. 5e, f, respectively (also see the Supplementary Movie 4). Binding occurred between the streptavidin-conjugated microparticle and *E. coli* conjugated with biotinylated-anti-*E. coli* antibody in a very short time (<100 ms)

as shown in Fig. 5e. Once bound, the microparticle–bacterium complex hopped away. Experimental demonstrations of selective binding and hopping of *S. flexneri* are shown in Supplementary Fig. 11 and Supplementary Movie 5. The binding process happens when the streptavidin-conjugated particle is in contact with the bacteria stained with biotin-labelled antibodies due to the strong interaction between streptavidin and biotin ($K_d = 10^{-15}$ M). To facilitate binding, the bacteria were saturated with biotin-labelled antibodies. The particle could easily be in contact with a single bacterium, bind with it and hop away together even with only a few bacteria trapped in the potential well.

**Discussion**

The binding efficiency was measured using our optofluidic lattice by counting the percentage of microparticles that hopped with bacterial cells (Table 2). For each experimental condition, 300 events were recorded to calculate the binding efficiency. The concentration of bacteria and streptavidin-coated silica microparticles were both $4 \times 10^7$ (cells) particles per ml. When the streptavidin-conjugated microparticle hit the bare *E. coli*, particle hopping was not observed, i.e., 100% *E. coli* remained being trapped in the potential well. When conjugated with the specific antibody, the binding efficiency of *E. coli* and *S. flexneri* to microparticles was measured as 93% and 87%, respectively. The two groups of bacteria and their specific antibodies showed similar binding affinities, which was confirmed by the SPR experiments (Supplementary Fig. 9). When measured with flow cytometry, *E. coli* showed higher cell count than *S. flexneri* in spite of the same concentration, suggesting higher binding efficiency of the anti-*E.coli* antibody. To measure the specificity of the two antibodies, we incubated *E. coli* with anti-*S. flexneri* antibody and vice versa. A total of 3% of *E. coli* conjugated with anti-*S. flexneri* antibody hopped away with microparticles, whereas no *S. flexneri* hopped with microparticles, suggesting the anti-*E. coli* antibody had higher specificity. In addition, the flow cytometry result suggested that the anti-*E. coli* antibody had slightly lower non-specific binding, which agreed with our result. The SPR could not detect any binding activity with non-specific antibodies (Supplementary Figs. 9 and 10). Meanwhile, the mean residence time for particles, either bound with bacteria or not, did not change significantly for all cases. Due to the higher laser power (400 mW) and smaller flow velocity (~50 µm s$^{-1}$) used in the bacteria experiments compared to the conditions (300 mW and 100 µm s$^{-1}$) used in pure particle experiments, the trapping positions for particles in the bacteria experiments had much shallower potential well compared to the adjacent potential well, which resulted in the mean residence time to be <0.2 s as shown in Supplementary Table 1. It is noted that the bacteria with rod shape should be aligned parallel to the flow direction (also the light propagating direction) in the laminar flow. Meanwhile, the bacteria with different shapes (diameter and length) experience different optical and fluidic forces[66,67], which only causes the distributions of trapping positions of bacteria in Fig. 5 and Supplementary Fig. 11. The working size range of the particle is from 500 nm to 2 µm. The optical scattering force on the 500 nm particle is about 3 pN, which is still much larger than the fluidic drag force (0.25 pN). However, a further decrease on the particle size will require a higher laser power (>1 W). On the other hand, when the particle size is larger than 2 µm, the particle may occupy more than one hotspot because the lateral distance between two hotspots (e.g., hotspots 1 and 4) is only 1.4 µm. It will disturb the optical field significantly, and the particle hopping may not occur.

Current measurement of binding efficiency relies on the manual counting of trapped bacteria and hopped microparticle–bacterium complexes, which is tedious and

laborious. The counting process can be improved by developing an image processing software for automatic bacteria tracking and counting. Moreover, the flow velocity used in the experiment is 50 $\mu$m s$^{-1}$, which results in a relatively low throughput. The flow velocity can be further improved by realising the optofluidic lattice with higher optical strength through the optimisation of optical lattice. Meanwhile, the culture and labelling of bacteria can be further integrated into a single optofluidic chip to facilitate the commercialisation of the chip.

Our optofluidic device provides a new way of detecting and screening of bacterial binding agents, and measurement of their binding efficiency at single cell level in a semi-quantitative manner based on the sophisticated but controllable multi-hopping phenomenon. Particle–particle interactions, which have long been overlooked, play critical roles in triggering multi-hopping in the lattice. As revealed in this study, the instantaneous and simultaneous multi-particle hopping process also modifies the conventional calculation of the single-particle residence time in potential wells. By counting the hopping rate of microparticle–bacterium complexes, the binding efficiency of bacteria and antibodies can be measured. Further tuning of the optofluidic lattice by changing the shape of the optical elements, the numerical aperture or the beam-waist position of the source beam may provide new degrees of freedom for diverse hopping mechanisms of functional and theoretical interest. Other methods for the generation of light with arbitrary shapes, such as optical holographic and optoelectronic techniques, could enable particles to hop and loop automatically with a constant flow rate in the future. Our studies offer an in-depth probe into long-ignored diverse multi-particle transitions between optical potential wells, open up new avenues for biomolecule interactions at single bacteria level, as well as inspire the future development of optical and Brownian motors in microfluidic systems.

## Methods

**Sample preparation**. *E. coli* (MCLAB) and *S. flexneri* (Sigma-Aldrich) were inoculated into a Luria-Bertani broth and nutrient broth, respectively, and cultured overnight in a 37 °C incubator. Bacteria cells were pelleted by centrifugation and then resuspended in phosphate-buffered saline (PBS). The cells were fixed with 4% paraformaldehyde, permeabilized with 0.1% Triton X-100 and blocked with 10% bovine serum albumin in PBS. Anti-*E. coli* antibody (Abcam, goat polyclonal antibody) and anti-*S. flexneri* antibody (Bio-Rad, mouse monoclonal antibody) were labelled by Mix-n-Stain biotin (Sigma-Aldrich). The labelled antibodies were purified with an ultrafiltration membrane column and centrifuged at 14,000 × g. Then, the bacteria cells were incubated with the biotin-labelled antibodies for 1 h at room temperature. The bacteria concentration was 4 × 10$^7$ cells per ml. Streptavidin-coated silica microparticles (Bangs laboratories, Inc.) with a diameter of 0.99 $\mu$m were suspended in PBS and the concentration was adjusted to 4 × 10$^7$ particles per ml.

**Optofluidic chip**. The optofluidic chip consisted of a microfluidic channel with three inlets and two outlets. The microchannel had dimensions of $L = 1000$ $\mu$m, $W = 80$ $\mu$m and $H = 100$ $\mu$m. The core flow was a particulate suspension and the side-flows were de-ionised water. A microlens was made from poly-dimethylsiloxane (PDMS) and placed at the edge of the microchannel (Supplementary Fig. 3). A discrete optical interference pattern was realised by irradiating light through the micro-quadrangular lens, which was coupled into the flow stream along the microchannel from the outlet to the inlet. A 3D simulation and a fluorescence image of the discrete interference pattern are shown in Supplementary Figs. 5 and 8, respectively.

**SPR binding assays**. SPR experiments were performed using a BIACORE T200 instrument (GE Healthcare) equipped with a CM5-S sensor chip. The surfaces of all flow cells were activated for 7 min with a 1:1 mixture of 0.1 M NHS (N-hydroxysuccinimide) and 0.1 M EDC (N-Ethyl-N′-(3-dimethylaminopropyl)car-bodiimide) at a flow rate of 10 $\mu$l min$^{-1}$. Neutravidin was captured on all surfaces to approximately 6000 RU. All the surfaces were then blocked with a 7 min injection of 1 M ethanolamine, pH 8.0. The ligands, biotinylated anti-*E. coli* antibodies (150 kDa) and anti-*S. flexneri* antibodies (150 kDa), at a concentration of 25 $\mu$g ml$^{-1}$ in PBS buffer were immobilised to the density of approximately 600 RU on flow cell 2 and 4 respectively; flow cell 1 and 3 was left blank to serve as a reference surface. PBS buffer blank or the analytes, *E. coli* cells (4 × 10$^7$ cells per ml)

and *S. flexneri* cells (4 × 10$^7$ cells per ml), in PBS buffer were injected over the all flow cells at a flow rate of 1 $\mu$l min$^{-1}$, for a contact time of 1200 s and dissociate for 5400 s. Data were collected at a rate of 1 Hz, and at a temperature of 25 °C. Sensorgrams were plotted with a common y-scale to allow direct comparison of the different samples. Measured results are shown in Supplementary Fig. 9.

**Flow cytometric analysis**. Alexa Fluor 568 (Thermo Fisher Scientific) was used to label antibodies for flow cytometric analysis. A total of 100 $\mu$l of antibody solution at 1 mg ml$^{-1}$ was incubated with reactive dye for 1 h at room temperature. A spin column was used to remove the unbound dye from the dye-conjugated antibodies. Then, 10 $\mu$l of dye-conjugated antibodies were added to 10 $\mu$l of bacteria with the concentration of 4 × 10$^7$ cells per ml, and incubated in the dark for 1 h at room temperature. After that, the samples were washed twice and diluted into 100 $\mu$l solution by PBS. *E. coli* and *S. flexneri* were both incubated with anti-*E. coli* antibodies and anti-*S. flexneri* antibodies, respectively. The four samples were analysed by flow cytometry (ImageStream X MarkII, Merck). For the fluorescence detection of Alexa Fluor 568, the excitation light was 561 nm and laser power was 100 mW. For the side scattering detection, the incident light was 785 nm and laser power was 6 mW. The samples were analysed by the flow cytometry with a stopping gate set at 10,000 events. The stained bacterial populations were determined by the region within the threshold of fluorescence intensity (1 × 10$^4$ to 3 × 10$^5$) and scattering intensity (5 × 10$^2$ to 2 × 10$^5$). Measured results are shown in Supplementary Fig. 10.

**Fabrication and experimental setup**. The optofluidic chip was fabricated using soft-lithography processes. First, photoresist-on-silicon masters of the chips were prepared by photolithography (Micro-Chem, SU-8) using transparent glass masks (CAD/Art Services Inc., Poway, CA, USA). Then, the microchannels and the structure of the micro-quadrangular lens were moulded using PDMS and sealed against flat PDMS slabs after oxygen plasma treatment.

An argon ion laser (532 nm, Laser Quantum, mpc 6000) was coupled to an optical fibre (cladding diameter: 125 $\mu$m, core diameter: 9 $\mu$m, NA: 0.12) and inserted into the fibre groove near the microlens. Particle hopping images were captured using an inverted optical microscope (TS 100 Eclipse, Nikon) through a charge-coupled device camera (Photron Fastcam SA3). Polystyrene particles were dispersed in water and injected into the microchannel using syringe pumps (Genie, Kent Scientific Corporation, CT, USA). The dye rhodamine 6 G with a concentration of 1 × 10$^{-7}$ mol l$^{-1}$ (excitation: 532 nm, emission: 550–590 nm) was added to visualise the ray trajectories in the microchannel. The confocal image of the optofluidic lattice was captured using a Leica TCS SP8. Experiments were performed under stable conditions at room temperature.

**Simulation of light field and optical forces**. The interference fields created by a Gaussian beam illuminating the micro-quadrangular lens were modelled using the finite-difference time-domain method in the commercial software Lumerical. The optical force was simulated simultaneously in Lumerical based on the Maxwell stress tensor, which can be expressed as[68,69]:

$$\mathbf{F} = \oint_S \langle T \rangle \cdot d\mathbf{S}, \tag{3}$$

where the integration is performed over a closed surface, and <T> is the time averaged Maxwell stress tensor, which can be expressed as:

$$\langle T \rangle = \frac{1}{2} \mathrm{Re} \left[ \varepsilon \mathbf{EE}^* + \mu \mathbf{HH}^* - \frac{1}{2} \left( \varepsilon |\mathbf{E}|^2 + \mu |\mathbf{H}|^2 I \right) \right], \tag{4}$$

where $\mathbf{EE}^*$ and $\mathbf{HH}^*$ denote the outer product of the fields, $I$ is the identify matrix, and $\varepsilon$ and $\mu$ are the relative permittivity and relative permeability of the medium, respectively. The forces $F_x$, $F_y$ and $F_z$ are the sum of all optical and fluidic forces in each direction. The potential energy profile was calculated as:

$$U_{x,y,z} = \int_{x,y,z} F_{x,y,z} dx, y, z. \tag{5}$$

**Calculation of mean first-passage time**. The MFPT characterises the mean residence times of the particles in the potential wells, and is expressed as[70]:

$$t_r = \frac{2\pi\zeta}{\sqrt{|U''(\alpha)||U''(\beta)|}} \exp\left(\frac{U(\beta) - U(\alpha)}{k_B T}\right), \tag{5}$$

where $\zeta$ is the fluidic damping constant with $\zeta = 6\pi\eta R$, where $\eta$ is the dynamic viscosity of the liquid and $R$ is the particle radius. $k_B$ is the Boltzmann constant, $T$ is the temperature and $U(\alpha)$ and $U(\beta)$ denote the potential energy at the valley point $\alpha$ and saddle point $\beta$, respectively. $U''(\alpha)$ and $U''(\beta)$ denote the second derivative of the potential energy profile at points $a$ and $b$, respectively.

**Data availability**. The data that support the findings of this study are available from the corresponding authors on reasonable request.

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

## Acknowledgements

This work was supported by the Singapore National Research Foundation under the Competitive Research Program (NRF-CRP13-2014-01), and under the Incentive for Research & Innovation Scheme (1102-IRIS-05-02) administered by PUB.

## Author contributions

Y.Z.S., C.-W.Q., and A.Q.L. jointly conceived the idea. Y.Z.S., S.X., Y.Z., J.B.Z., W.S., J.H.W., T.N.C., Z.C.Y., Y.L.H., B.L., K.W., P.H.Y., D.P.T., and C.-W.Q. performed the numerical simulations and theoretical analysis. Y.Z.S, S.X., and L.K.C. did the fabrication and experiments of particle hopping, biomolecule binding and flow cytometry. A.L. and S.H.L. did the SPR experiments. S.X., Y.Z.S., Y.Z., C.-W.Q., Y.-Y.C., L.K.C., T.H.Z., and A.Q.L. prepared the manuscript. S.X., Y. Z., C.-W.Q., and A.Q.L. supervised and coordinated all the work. All authors commented on the manuscript.
