## [Peer Review File · Nature Communications]

Reviewers' comments:

Reviewer #1 (Remarks to the Author):

According to the paper, the claims are two-fold: 1. Basic study of multi-particle hopping mechanism due to mechanical interactions. The authors have shown to use optofluidic lattice to observe, characterize and analyze the long-neglected hopping mechanisms of multi-particles between hotspots. 2. Challenging application of one of the mechanisms, i.e., aggregate-particle hopping for bio-molecule binding analysis, as a new way for single-cell level screening of binding efficiency. The authors have demonstrated to measure the specificity and affinity of antibody binding to single bacterial cells in a semi-quantitative manner. To the best of my knowledge, the two claims are fair and important.

The group led by the corresponding author has long been working on optofluidics and the work reported in this paper was another smart piece that discovered the hopping mechanisms of multiple particles. Though the particle-particle interactions happen naturally like snookers in our macro-world, to study the same phenomena of micro-sized particles in the context of microfluidics-setting optical trapping is non-trivial. It requires solid technological background and rich experience in optofluidics. Perhaps, the field ignores the study of hopping of multiple particles between hotspots partially because of lack of the expertise and insights. In this sense, I would say this work brought technology-enabled new findings on multi-particle hopping and I believe that the basic study will bring broad impact and value in applied physics including optics and fluidics.

By the same token, the application of one of the identified mechanisms into bio-molecule binding assay established a new paradigm for tackling the challenging and key issue of specificity and affinity associated with single bacterial cells. Though the demonstrated assay is far below advanced from the long-developed mature methods such as ELISA and SPR, it provides an alternative way worth exploring to further increase the throughput and automation level. In summary, this application beautifully exemplified the usefulness of the basic study and finding of the hopping mechanisms.

Having said that, I have the following concerns:

1. It is very widely useful if the particle can be steered to hop in a particular trajectory. The authors have provided Fig. S1 and S2 as supplementary information to explain how to maneuver one particle to hop between the four hotspots. In theory, this sounds acceptable. However, to reinforce the statement, I would recommend the authors to provide an additional video clip to demonstrate the maneuverability. I believe this would benefit our readers in two aspects: i) the movement model is valid, ii) the potential can be more than expected. Overall, this demonstration would strengthen the impact.
2. Fig. 1c plots the hotspots as a semi-ellipsoid. This may be misleading. I imagine that the hotspots are an ellipsoid. The semi-ellipsoid I guess is to show the cross-section, with light intensity. In addition, the definition of hotspot is not clear-cut. Is it a region, like the whole 3D space contained by the ellipsoid? Or is it simply the centroid (which has the highest light intensity) of the ellipsoid? I suggest the authors single out a hotspot in the plotting. Plus, clarify the relative positions between a hotspot and the potential well.
3. Fig. 1def the first row is to outline the geometric locations of multiple particles for 2D hopping. However, I found it quite confusing to read the vertical plane with the red scattering light spot. The two orthogonal planes do not seem to have the corresponding light intensity patterns consistent with the illustration in Fig. 1c. This is a bit disturbing in understanding. Similarly, the second row of Fig. 1def, I understand, is to plot the energy profile of the hotspots. However, neither the main text nor the figure caption explains something about it. The readers need to figure it out from the information introduced later.

4. Fig. 2bc, the caption is wrongly stated. Plus, in Fig. 2ab, the horizontal plane, the label '50' turns to be redundantly wrong. So are Fig. 3/5, and the related video clips.
5. Line 264, the parameter 0.7 (μm), need to be defined as generally as the experimental settings.
6. According to line 248, two particles tend to aggregate occasionally. So it is critical to comment on the biological experiment, how to control the experimental setup such that the conjugation can be formed easily to facilitate the throughput. For example, how many bacterial cells could be trapped on the hotspots before aggregation could happen to move one of the cells? How to control the microparticles coated with antibodies to facilitate the aggregations?
7. The paper provides a measure of the binding efficiency for bacterial cells using the aggregation mechanism. It is important to provide data and comments on the sample number, measurement repeatability and error. It is also valuable to provide some discussion on the aspects that could be improved for the proposed binding efficiency measurement method.
8. Fig. S2a, in the figure, there are U1U2U3U4 and dashed lines. I guess these are the boundaries for the four potential wells? There should be some words to explain the labels. Plus, in the figure caption, U1(z=70um), U2(z=80um), U3(z=90um), U4(z=100um) seem wrong to me. Fig. S5, suggest to add some words in the caption to describe the trajectory starting point and end point. Currently, all trajectories are plotted in white, leading to poor legibility about the difference among the trajectories, if any, for different particles. Fig. S6, the labelling of angle theta is not complete.
9. Fig. 3cd, Fig. 4d, the figure for energy profile is a bit confusing. Where is the location in the hotspot corresponding to $x=0$? For these profiles, $y=?$ $z=?$
10. The abstract reads a bit long. It can be shortened.

Reviewer #2 (Remarks to the Author):

The manuscript by Shi and coworkers describes controlled microparticle and bacteria collision experiments performed in an 'optofluidic' chip consisting of a conventional microfluidic system coupled to a microlens that produces a non-Gaussian interference pattern in the channel that is capable of weakly trapping small refractile objects. The central claim of the paper is that this instrument allows the semi-quantitative assessment of specific and non-specific binding affinity to whole, single bacteria. Overall, the work appears to be carefully done, and the results of the study are statistically compatible with the authors' claims. This reviewer found the description of the potential impact of such an assay in its current form to be overstated....what functionality does it allow that is not possible with SPR + FACS type approach? Nevertheless, I could readily imagine variations on the experiments presented that would be of greater interest. It also seems likely that the authors had similar long-term motivations in mind, but that this was not clearly conveyed in the current manuscript. On this basis, I think the manuscript should be ultimately acceptable in Nature Communications if some minor concerns could be addressed.

--The cool factor of this work was only clear upon repeated reading. The introduction was very difficult to follow and seemed overly long (the latter issue might be better addressed by an Editor). This seems to be both a function of English language usage and a subtle mis/over-statement of the claims. From my point of view, the excitement of this method is the ability to (statistically) manipulate multiple microscopic objects in buffer, collide them and shift them between wells in a semi-controlled fashion (by modulating laser power and flow rate). This is combined in a device that is (presumably) mass-producible (wetted parts), conducive to high performance optical microscopy and semi-automated image analysis and operation. The 'optical' feature is that as a non-contact method it is free of fouling and non-specific binding between the test objects and the 'hot spot' manipulators. I thought that figure S1 showed this degree of control quite nicely.

--I am loathe to instruct non-native English speakers about their usage....but the authors would do well to hire a technical translator, or if they have already to hire a better one. One example..change 'captivate' to 'consider'. The editors at NPG will presumably also be of use should

the paper be accepted.

--I like that the paper performed extensive modeling and validation of the optical field. I would consider the force produced by Eq 3 however to only be valid for a Rayleigh particle, and would expect the forces on a Mie particle to be more complicated (but qualitatively similar). Of course, as mentioned there are also non-additive optical interactions between particles. In the face of this, it is hard to take the potential curves and energy values (and comparisons to Kramers theory) too seriously. Perhaps the authors can address these concerns in a revised manuscript. And perhaps some of the discussion of Kramers phenomena can be deprecated and/or moved to the Supplementary Material. In my opinion, the potential model is important to illuminate the behavior of the particles and design of the experiments, but quantitative closure is not required.

Reviewer #3 (Remarks to the Author):

The authors investigate the use of an optofluidic lattice for use in quantifying bacterial binding to surface-modified beads. As the authors point out, there has been considerable work over the years in particle hopping between potential wells using optical trapping methods. Most of this work has been with single particles jumping between wells and the contribution here, to some degree, is to extend to multiparticle systems where interparticle interactions play a significant role. Key in their approach is a balance of multiple forces, including fluid drag, optical gradient and scattering forces, as well as Brownian motion. Superimposed on all of these are the forces imparted by flowing particle and cell collisions. While an interesting demonstration, I believe the approach is not suited for the throughputs required to make it a useful, quantitative method. As a result, I feel the manuscript is unsuitable for publication in Nature Communications.

Specific Comments

1) The authors' motivation that comparable screening technologies are time consuming and require a large sample needs additional support. In practice, how do the authors propose a micro-optofluidic antibody screening device would work and with what amount of sample? Well established methods such as ELISA and flow cytometry can perform closely-related measurements at rates many orders of magnitude higher than the demonstrated single cell technique. Is the motivation for performing their analysis in an optofluidic device to obtain high throughput single-cell analysis or something else?

2) The approach here requires the balancing of many forces including those which can be hard to control in practice. Drag forces in particular can be difficult as they depend on flow rate (which can fluctuate depending on pumping method), distance from channel walls (which tend to be close in confining microfluidic geometries making variations significant), and particle geometry/orientation (which is very different for bacteria vs. beads). As a result, it is difficult to envision how the approach could be implemented in practice or whether it could be performed at rates high enough to get necessary statistics (and it is not clear how long it took to get the measurements needed for the data that is presented). This concern is exacerbated somewhat by a lack of any discussion of relative force magnitudes, sensitivity analysis, or even error bars/number of measurements in Tables 1, 2, and S1, all while the authors do note that the approach requires "delicately balancing the preceding forces..." on pg. 7.

3) Unlike the title, the measurement of binding efficiency does not appear to be quantitative. The authors point this out themselves by referring to the technique as "semi-quantitative" in the conclusion.

5) The manuscript requires significant editorial work – sections of the manuscript were very

difficult to understand.

6) Ref 3 seems misplaced.

Reviewer #4 (Remarks to the Author):

The paper propose a novel optofluidic chip for device based on nano-optofluidic lattice enables to manipulate individual bacterial cells in the flow stream. The approach represent an evolution optical chromatography and an enhancement respect to the previous paper, nor reported in the references, of the same Authors:

Y. Z. Shi, S. Xiong, L. K. Chin, Y. Yang, J. B. Zhang, W. Ser, J. H. Wu, T. N. Chen, Z. C. Yang, Y. L. Hao, B. Liedberg, P. H. Yap, Y. Zhang and A. Q. Liu, "High-resolution and multi-range particle separation by microscopic vibration in an optofluidic chip", *Lab Chip*, 2017, 17, 2443-2450.

The use of multiple hotspots permits to add very powerful functionalities like particle bypassing, collision and aggregation. The experimental results on the binding affinity are very promising.

There are some questions that should be addressed:

In the above paper the Authors use a micro-quadrangular lens in order to obtain a quasi-Bessel beam. In this paper the same micro-quadrangular lens is used in order to obtain the optical interference pattern with four hotspot.

The optical interference pattern forms an array of non-uniform hotspots. The role of non-uniformity of the intensity between the four hotspots should be addressed in relationship to the three hopping mechanisms (i.e., particle bypassing, collision and aggregation).

The Authors shows results with particle and bacteria with size of about 1 micron. The particle the working size range of the proposed device should be analysed.

Particle hopping triggered by lateral drag force in x-direction should be deeply discussed. The source of the lateral drag force should be explained (asymmetric hydrofocussing?)

The reference list on optical chromatography should be improved

Manuscript ID: NCOMMS-17-21398

Paper title: **Sculpting Nanoparticle Snooker for Single-bacteria-level Screening and Direct Binding-efficiency Quantification**

Authors: **Y. Z. Shi, S. Xiong, Y. Zhang, L. K. Chin, Y. -Y. Chen, J. B. Zhang, T. H. Zhang, W. Ser, A. Larson, L. S. Hoi, J. H. Wu, T. N. Chen, Z. C. Yang, Y. L. Hao, B. Liedberg, P. H. Yap, D. P. Tsai, C.-W. Qiu, and A. Q. Liu**

Reply to Reviewer 1

We are grateful to the Reviewer for the constructive comments and are delighted that the Reviewer is interested in our results and recommends the publication of this manuscript. We are happy to address all the comments.

Comment: *It is very widely useful if the particle can be steered to hop in a particular trajectory. The authors have provided Fig. S1 and S2 as supplementary information to explain how to maneuver one particle to hop between the four hotspots. In theory, this sounds acceptable. However, to reinforce the statement, I would recommend the authors to provide an additional video clip to demonstrate the maneuverability. I believe this would benefit our readers in two aspects: i) the movement model is valid, ii) the potential can be more than expected. Overall, this demonstration would strengthen the impact.*

Reply: As suggested by the reviewer, an additional video is added as Supplementary Movie 7 to demonstrate the particle can be steered to move around a hopping loop with four hotspots, which is controlled by optical and fluidic forces. The controllable optical hopping loop is presented in Figure S3.

Comment: *Fig. 1c plots the hotspots as a semi-ellipsoid. This may be misleading. I imagine that the hotspots are an ellipsoid. The semi-ellipsoid I guess is to show the cross-section, with light intensity. In addition, the definition of hotspot is not clear-cut. Is it a region, like the whole 3D space contained by the ellipsoid? Or is it simply the centroid (which has the highest light intensity) of the ellipsoid? I suggest the authors single out a hotspot in the plotting. Plus, clarify the relative positions between a hotspot and the potential well.*

Reply: As suggested by the reviewer, in the revised manuscript, Figure 1c is modified. The intensity profile of the hotspot is changed to an ellipsoid and a hotspot is singled out. In addition, the relative position between a hotspot and the potential well is clarified as “A potential well is created accordingly with the center point at $P_1 (x_1, y_1, z_1)$. The injection point of the laser in the microchannel is set as the zero point $P_0 (0, 0, 0)$ as shown in Figure 1a. Since the hopping between two adjacent hotspots happens along the x direction, the potential energy profile is plotted along the x axis at $y = y_1$ and $z = z_1$ in the bottom row of Figure 1d-f.” in line 131 page 6.

Figure 1c Illustration of the realization of controllable particle hopping loop around the ellipsoid hotspots.

Comment: *Fig. 1def* the first row is to outline the geometric locations of multiple particles for 2D hopping. However, I found it quite confusing to read the vertical plane with the red scattering light spot. The two orthogonal planes do not seem to have the corresponding light intensity patterns consistent with the illustration in Fig. 1c. This is a bit disturbing in understanding. Similarly, the second row of Fig. 1def, I understand, is to plot the energy profile of the hotspots. However, neither the main text nor the figure caption explains something about it. The readers need to figure it out from the information introduced later.

Reply: As pointed out by the reviewer, Figure 1c is an illustration showing the overview of the optical field, while Figure 1d-f shows the simulated light intensity patterns. In Figure 1d-f, the intensity profiles in y - z plane (vertical) are slightly shifted from the central line ($x = 0$) to have a better view of the particle trajectories. The figure caption of Figure 1d-f is revised as “Illustration of the simulated light intensity (top row) and potential energy (bottom row) profiles of particle hopping triggered by three different mechanisms: (d) particle bypassing, (e) snookering-like particle collision and (f) particle aggregation. The intensity profiles in y - z plane (vertical) are slightly shifted from the central line ($x = 0$) to have a better view of the particle trajectories.” In addition, in the revised manuscript, the energy profile is discussed as “Since the hopping between two adjacent hotspots happens along the x direction, the potential energy profile is plotted along the x axis at $y = y_1, z = z_1$ in the bottom row of Figure 1d-f.” in line 133 page 6.

Comment: *Fig. 2bc, the caption is wrongly stated. Plus, in Fig. 2ab, the horizontal plane, the label '50' turns to be redundantly wrong. So are Fig. 3/5, and the related video clips.*

Reply: As pointed out by the reviewer, in the revised manuscript, the caption of Figure 2 is revised as “Illustration of (a) the green particle being trapped in the extra trapping position and (b) hopping to the adjacent potential well. Optical intensity profiles at the x - z plane ($y = 0$) and y - z plane ($x = 0$) are plotted. (c) Force analysis on the 1- μm polystyrene particle along the z direction when $x = 0$, $y = 0$. Blue line represents the drag force. Red and green lines represent the optical forces on red and green particles, respectively. (d) Projection of the green particle trajectory on the x -axis. (e) Experimental demonstration of the 1- μm particle hopping induced by the pre-trapped particle.” The label of z -axis of Figures 2a-b, 3a-b and 5a-d are revised.

Comment: *Line 264, the parameter 0.7 (μm), need to be defined as generally as the experimental settings.*

Reply: As suggested by the reviewer, in the revised manuscript, the parameter is redefined as “The critical position b can be expressed as $b = H/2 - 0.5R\sin\theta$, where H is the distance between the centers of two adjacent hotspots in x -direction and R is the radius of red particle. Similarly, the critical position a can be expressed as $a = -b = -H/2 + 0.5R\sin\theta$. In our experiments, H is 1.4 μm .” in line 235 page 11.

Comment: *According to line 248, two particles tend to aggregate occasionally. So it is critical to comment on the biological experiment, how to control the experimental setup such that the conjugation can be formed easily to facilitate the throughput. For example, how many bacterial cells could be trapped on the hotspots before aggregation could happen to move one of the cells? How to control the microparticles coated with antibodies to facilitate the aggregations?*

Reply: As suggested by the reviewer, in the revised manuscript, the control of the biological experiment is discussed as “The binding process occurs when the streptavidin-conjugated particle is in contact with the bacteria conjugated with biotin-labelled antibodies due to the strong interaction between streptavidin and biotin ($K_d = 10^{-15}\text{M}$). To facilitate the binding, bacteria are saturated with biotin-labelled antibodies. The particle could easily be in contact with a single bacterium, bind with it and hop away together even with only a few bacteria being trapped in the potential well.” in Line 306 Page 14.

Comment: *The paper provides a measure of the binding efficiency for bacterial cells using the aggregation mechanism. It is important to provide data and comments on the sample number, measurement repeatability and error. It is also valuable to provide some discussion on the aspects that could be improved for the proposed binding efficiency measurement method.*

Reply: As suggested by the reviewer, more detailed experimental data are addressed as “The binding efficiency was measured using our optofluidic lattice by counting the percentage of

microparticles that hopped with bacterial cells (Table 2). For each experimental condition, 300 events were recorded to calculate the binding efficiency. The concentration of bacteria and streptavidin coated silica microparticles were both 4×10^7 (cells) particles/mL.” in Line 312 Page 14.

Further improvement on the binding efficiency measurement method is discussed as “Current approach to measure the binding efficiency relies on the manual counting of trapped bacteria and hopped microparticle-bacterium complexes, which is tedious and laborious. The counting process can be improved by developing an image processing software for automatic bacteria tracking and counting. Moreover, the flow velocity used in the experiment is 50 $\mu\text{m/s}$, which has relatively low throughput. The flow velocity can be further improved by realizing the optofluidic lattice with higher optical strength through the optimization of optical lattice.” in Line 344 Page 16.

Comment: *Fig. S2a, in the figure, there are $U_1U_2U_3U_4$ and dashed lines. I guess these are the boundaries for the four potential wells? There should be some words to explain the labels. Plus, in the figure caption, $U_1(z=70\mu\text{m})$, $U_2(z=80\mu\text{m})$, $U_3(z=90\mu\text{m})$, $U_4(z=100\mu\text{m})$ seem wrong to me. Fig. S5, suggest to add some words in the caption to describe the trajectory starting point and end point. Currently, all trajectories are plotted in white, leading to poor legibility about the difference among the trajectories, if any, for different particles. Fig. S6, the labelling of angle θ is not complete.*

Reply: As pointed out by the reviewer, the dashed lines represent the positions ($z = 50, 45, 40$ and $35 \mu\text{m}$) of the four potential wells (U_1, U_2, U_3 and U_4) when the particle hops from hotspot 3 to 4, which is explained in the revised manuscript as “1- μm particle (red) is driven by the drag force and hop from hotspot 3 to 4 subsequently through four potential wells (dashed lines) labelled with $U_1(z = 50 \mu\text{m})$, $U_2(z = 45 \mu\text{m})$, $U_3(z = 40 \mu\text{m})$ and $U_4(z = 35 \mu\text{m})$. The intensity pattern is rescaled for a clear illustration.” in the caption of Figure S2. In addition, the scale and ticks are labeled on the x, z -axis in Figure S2a.

In the revised Fig. S6 (original Fig. S5), the particle trajectories are presented in different colours for better visualisation. The starting point and end point are discussed as “The dots represent the starting points of the particles when $t = 0$. The trajectories ending time of the particles in (a) and (b) are 0.3 and 3 seconds, respectively. (a) In the x - z planes, the starting points of the particles are at $z = 5 \mu\text{m}$, which are on the left edge of hotspot 1. x is from -1.25 to $1.25 \mu\text{m}$. After $t = 0.2 \text{ s}$, all particles are attracted to the center line of the hotspot ($x = 0$) and pushed to the right edge of the hotspot. (b) In the y - z planes, the starting points of the particles are also at $z = 5 \mu\text{m}$. y is from -60 to $60 \mu\text{m}$. The strong optical force drives all particles in the hotspot ($-20 \mu\text{m} < y < 20 \mu\text{m}$) to move towards the stable trapping point ($y = 0$ and $z = 20 \mu\text{m}$), whereby the optical force balances the fluidic drag force. While particles beyond this range ($-20 \mu\text{m} < y < 20 \mu\text{m}$) are flushed away by the drag force.” in the caption of Fig. S6.

In the revised Fig. S7 (original Fig. S6), the label of angle θ is amended.

Figure S6 | Simulation of particle trajectories in (a) x - z and (b) y - z planes.

Figure S7 | Illustration of the contact angle and the vertical distance between the two particles after collision.

Comment: *Fig. 3cd, Fig. 4d, the figure for energy profile is a bit confusing. Where is the location in the hotspot corresponding to $x=0$? For these profiles, $y=? z=?$*

Reply: As pointed out by the reviewer, the energy profiles for Figures 3c-d and 4c are all plotted along the x axis at $y = 0$ and $z = 20 \mu\text{m}$ because the particle is pre-trapped at the point $P_1 (0, 0, 20)$ and the particle interaction happens along the x -direction. In the revised manuscript, the description on the energy profiles is added as

“Before the collision, the red particle is trapped at the valley point of the potential well at point α (Figure 3c), i.e. at $x = 0, y = 0$ and $z = 20 \mu\text{m}$.” in Line 197 Page 9;

“The energy profile of the potential wells (c) before and (d) after the particle collision, plotted along x -axis at $y = 0$ and $z = 20 \mu\text{m}$.” in the Caption of Figure 3;

“The profiles of the potential wells located at particle pre-trapping position ($y = 0, z = 20 \mu\text{m}$) with different contact angles are plotted in Figure 4c.” in Line 239 Page 11; and

“Potential wells of total force F_t plotted along x -axis at $y = 0$ and $z = 20 \mu\text{m}$ with different contact angles.” in the Caption of Figure 4.

Comment: *The abstract reads a bit long. It can be shortened.*

Reply: As pointed by the reviewer, in the revised version, the abstract is shortened as “Isolated particle drifting and binding in tailored optical landscapes have been well addressed and used a versatile paradigm for various biomedical applications. However, the rich degrees of freedom of multi-particle “Snooker” in an optical lattice is unfortunately hurdled by the lack of implementation system. By judiciously steering the optofluidic lattice, we demonstrate the precise maneuver of sophisticated yet controllable Snooker-like events and functions such as bypassing, aggregation and multi-hopping. The mean residence time of a single particle was found considerably reduced from 7 s, as predicted by the longstanding Kramer’s theory, to 0.6 s, owing to the mechanical interactions among the aggregated particles. The sculpted nano-optofluidic lattice also enables the promising single-bacteria-level screening of biological binding agents such as antibodies through particle-enabled bacteria hopping and sorting. The binding-efficiency of antibodies could be determined directly, selectively, quantitatively and efficiently. This work enriches the fundamental mechanisms of particle kinetics and offers new possibilities for probing and utilizing unprecedented biomolecule interactions at single-bacteria level.” in Page 2.

In summary, we have addressed all comments of the reviewer. The manuscript and supplementary materials have been carefully corrected.

Reply to Reviewer 2

We are grateful to the Reviewer for the constructive comments and are delighted that the Reviewer is interested in our results and recommends the publication of this manuscript. We are happy to address all the comments.

Comment 1: *The manuscript by Shi and coworkers describes controlled microparticle and bacteria collision experiments performed in an 'optofluidic' chip consisting of a conventional microfluidic system coupled to a microlens that produces a non-Gaussian interference pattern in the channel that is capable of weakly trapping small refractile objects. The central claim of the paper is that this instrument allows the semi-quantitative assessment of specific and non-specific binding affinity to whole, single bacteria. Overall, the work appears to be carefully done, and the results of the study are statistically compatible with the authors' claims. This reviewer found the description of the potential impact of such an assay in its current form to be overstated....what functionality does it allow that is not possible with SPR + FACS type approach? Nevertheless, I could readily imagine variations on the experiments presented that would be of greater interest. It also seems likely that the authors had similar long-term motivations in mind, but that this was not clearly conveyed in the current manuscript. On this basis, I think the manuscript should be ultimately acceptable in Nature Communications if some minor concerns could be addressed.*

Reply: As pointed out by the reviewer, in the revised manuscript, the significant novelty of our work and comparison with conventional methods (e.g. ELISA, SPR, FACS) are discussed as “Apart from particle hopping, the optical potential wells enable the trapping of individual bacterial cells in the microfluidic channel to screen biological binding agents, such as antibody [47, 48], peptide or aptamer [49–51], and evaluate the binding affinity and specificity at single bacteria level. These binding agents play a crucial role in the pathogen recognition and inhibition in diagnostics and therapeutics of infectious disease [52–57]. Recently, single bacterial isolation and detection has been an emerging technique because the heterogeneity between individual bacteria cannot be revealed by conventional bulk approaches [58, 59]. Therefore, it is highly desired to screen the biological binding agents, such as antibodies, at single-cell level to reveal new insights of complex biological interactions. However, such needs are not fulfilled by conventional binding assays such as precipitation assay [60], agglutination assay [61], enzyme-linked immunosorbent assay (ELISA) [62, 63], surface plasma resonance (SPR) [64–67], Western blot [68] and fluorescence-activated cell sorting (FACS) [69]. ELISA generates colorimetric or fluorescent signals and evaluates the quantity of sample input by interpolating against a standard curve. SPR determines the dissociation constant (Kd), which is a measurement

of the binding affinity. However, these two assays require a relatively large number of bacterial cells and delicate processing of multiple step reactions. The bacteria loads in clinical samples are often too low for direct detection. A lengthy bacterial culture would be required to enrich the sample for detection, which takes half a day to one day. FACS is a technique capable of counting the binding efficiency between stained bacterial cells and microparticles. However, the extra staining process may interfere with the downstream assays and the microparticles are likely to affect the signal readout. Furthermore, multiple cells probably bind to the same microparticle and be counted as one. With our single-cell approach, we aim to work directly with clinical samples, which have low bacterial load, and shorten the turnaround time for potential diagnostic applications.” in Line 84 Page 4.

Comment 2: *The cool factor of this work was only clear upon repeated reading. The introduction was very difficult to follow and seemed overly long (the latter issue might be better addressed by an Editor). This seems to be both a function of English language usage and a subtle mis/over-statement of the claims. From my point of view, the excitement of this method is the ability to (statistically) manipulate multiple microscopic objects in buffer, collide them and shift them between wells in a semi-controlled fashion (by modulating laser power and flow rate). This is combined in a device that is (presumably) mass-producible (wetted parts), conducive to high performance optical microscopy and semi-automated image analysis and operation. The 'optical' feature is that as a non-contact method it is free of fouling and non-specific binding between the test objects and the 'hot spot' manipulators. I thought that figure S1 showed this degree of control quite nicely.*

Reply: As pointed out by the reviewer, in the revised manuscript, we have shortened the introduction and emphasized our claims clearly.

Comment 3: *I am loathe to instruct non-native English speakers about their usage....but the authors would do well to hire a technical translator, or if they have already to hire a better one. One example..change 'captivate' to 'consider'. The editors at NPG will presumably also be of use should the paper be accepted.*

Reply: As suggested by the reviewer, we have engaged a professional native English speaker to polish our paper. The English in the main text and Supplementary material has been carefully revised and improved.

Comment 4: *I like that the paper performed extensive modeling and validation of the optical field. I would consider the force produced by Eq 3 however to only be valid for a Rayleigh particle, and would expect the forces on a Mie particle to be more complicated (but qualitatively similar). Of course, as mentioned there are also non-additive optical interactions between particles. In the face of this, it is hard to take the potential curves and energy values (and comparisons to Kramers theory) too seriously. Perhaps the authors can address these concerns in a revised manuscript. And perhaps some of the discussion of Kramers phenomena can be deprecated and/or moved to the Supplementary Material. In my opinion, the potential model is important to illuminate the behavior of the particles and design of the experiments, but quantitative closure is not required.*

Reply: As pointed out by the reviewer, Eq. 3 has been changed to a general formula of Maxwell stress tensor and added in the revised manuscript as “The optical force was simulated simultaneously in Lumerical based on the Maxwell stress tensor, which can be expressed as [89, 90]

$$\mathbf{F} = \oint_S \langle T \rangle \cdot d\mathbf{S} \quad (3)$$

where the integration is performed over a closed surface, and $\langle T \rangle$ is the time averaged Maxwell stress tensor which can be expressed as

$$\langle T \rangle = \frac{1}{2} \text{Re} \left[\epsilon \mathbf{E} \mathbf{E}^* + \mu \mathbf{H} \mathbf{H}^* - \frac{1}{2} (\epsilon |\mathbf{E}|^2 + \mu |\mathbf{H}|^2 I) \right] \quad (4)$$

where $\mathbf{E} \mathbf{E}^*$ and $\mathbf{H} \mathbf{H}^*$ denote the outer product of the fields, I is the identify matrix, and ϵ and μ are the relative permittivity and relative permeability of the medium, respectively.” in Line 450 Page 20.

Our calculation is performed using commercial FDTD software Lumerical, which is based on the rigorous Maxwell stress tensor applicable to all kinds of particles. Therefore, the calculation of the potential well is also rigorous. Besides, we calculate the residence time of green particle in the potential well with a center at the extra position ($P_1(x_1, y_1, z_1)$) along the x -direction in Table 1. The red particle has no influence on the hopping progress or the potential wells along the hopping trajectory in the x -direction after it is pre-trapped. The experimentally observed mean residence time is in good agreement with the calculation using Kramer’s theory.

In summary, we have addressed all comments of the reviewer. The manuscript and supplementary materials have been carefully corrected.

Reply to Reviewer 3

We are grateful to the Reviewer for the constructive comments and are delighted that the Reviewer is interested in our results. We are happy to address all the comments.

Comment 1: *The authors' motivation that comparable screening technologies are time consuming and require a large sample needs additional support. In practice, how do the authors propose a micro-optofluidic antibody screening device would work and with what amount of sample? Well established methods such as ELISA and flow cytometry can perform closely-related measurements at rates many orders of magnitude higher than the demonstrated single cell technique. Is the motivation for performing their analysis in an optofluidic device to obtain high throughput single-cell analysis or something else?*

Reply: As pointed out by the reviewer, in the revised manuscript, the significant novelty of our work and comparison with conventional methods are discussed as “Apart from particle hopping, the optical potential wells enable the trapping of individual bacterial cells in the microfluidic channel to screen biological binding agents, such as antibody [47, 48], peptide or aptamer [49–51], and evaluate the binding affinity and specificity at single bacteria level. These binding agents play a crucial role in the pathogen recognition and inhibition in diagnostics and therapeutics of infectious disease [52–57]. Recently, single bacterial isolation and detection has been an emerging technique because the heterogeneity between individual bacteria cannot be revealed by conventional bulk approaches [58, 59]. Therefore, it is highly desired to screen the biological binding agents, such as antibodies, at single-cell level to reveal new insights of complex biological interactions. However, such needs are not fulfilled by conventional binding assays such as precipitation assay [60], agglutination assay [61], enzyme-linked immunosorbent assay (ELISA) [62, 63], surface plasma resonance (SPR) [64–67], Western blot [68] and fluorescence-activated cell sorting (FACS) [69]. ELISA generates colorimetric or fluorescent signals and evaluates the quantity of sample input by interpolating against a standard curve. SPR determines the dissociation constant (K_d), which is a measurement of the binding affinity. However, these two assays require a relatively large number of bacterial cells and delicate processing of multiple step reactions. The bacteria loads in clinical samples are often too low for direct detection. A lengthy bacterial culture would be required to enrich the sample for detection, which takes half a day to one day. FACS is a technique capable of counting the binding efficiency between stained bacterial cells and microparticles. However, the extra staining process may interfere with the downstream assays and the microparticles are likely to affect the signal readout. Furthermore, multiple cells probably bind to the same microparticle and be counted as

one. With our single-cell approach, we aim to work directly with clinical samples, which have low bacterial load, and shorten the turnaround time for potential diagnostic applications.” In Line 84 Page 4.

Comment 2: *The approach here requires the balancing of many forces including those which can be hard to control in practice. Drag forces in particular can be difficult as they depend on flow rate (which can fluctuate depending on pumping method), distance from channel walls (which tend to be close in confining microfluidic geometries making variations significant), and particle geometry/orientation (which is very different for bacteria vs. beads). As a result, it is difficult to envision how the approach could be implemented in practice or whether it could be performed at rates high enough to get necessary statistics (and it is not clear how long it took to get the measurements needed for the data that is presented). This concern is exacerbated somewhat by a lack of any discussion of relative force magnitudes, sensitivity analysis, or even error bars/number of measurements in Tables 1, 2, and S1, all while the authors do note that the approach requires “delicately balancing the preceding forces...” on pg. 7.*

Reply: As pointed out by the reviewer, the control of optical and fluidic forces is the key point of our proposed technique. The optical force is controlled by the laser power. The fluidic drag force is easily controlled by precision pumps which can achieve extreme low flow rate, e.g., < 1 nL/min (velocity < 2 $\mu\text{m/s}$) with very stable linear tuning. Moreover, even if the slight vibration from the environment may cause the deviation of the trapping positions in the flow stream, it would not affect the experiment at all. Since the optofluidic chip used hydrodynamic focusing, the optical field and particle interactions occurs in the center flow stream of the microchannel, whereby the wall effect can be ignored. In the experiment, the bacteria were trapped in a line in the hotspot 1 due to the bacterial size distribution (see Figure 5 and Figure S10).

In the revised manuscript, more experimental details are discussed as

“The experimental residence time is obtained based on 20 experimental events, in which the time ranges from 5 to 9 s.” in Line 175 Page 8;

“The magnitude of the maximum optical forces on the 1- μm polystyrene particle and *E. coli/S. flexneri* are approximately 20 and 1 pN, respectively. The drag force, which is proportional to the diameter of particle, is approximately 0.5 pN. Therefore, *E. coli* can be trapped on the hotspot 1 while the optical scattering force pushes the particle to the edge of the hotspot.” in Line 285 Page 13;

“For each experimental condition, 300 events were recorded to calculate the binding efficiency.” in Line 313 Page 14; and

“Mean residence time for different bacteria and antibodies (Sample size: 300 events).” in the caption of Table S1.

Comment 3: *Unlike the title, the measurement of binding efficiency does not appear to be quantitative. The authors point this out themselves by referring to the technique as “semi-quantitative” in the conclusion.*

Reply: As pointed by the reviewer, this technique is only semi-quantitative because it does not measure the absolute binding constants in terms of K_d . However, it can measure the relative binding efficiency with low sample load and shorter turnaround time for potential diagnostic application. In the revised manuscript, the title is revised as “Sculpting Nanoparticle Snooker for Single-bacteria-level Screening and Direct Binding-efficiency Measurement”.

Comment 4: *The manuscript requires significant editorial work – sections of the manuscript were very difficult to understand.*

Reply: As pointed by the reviewer, we have engaged a professional native English speaker to improve our paper for better understanding.

Comment 5: *Ref 3 seems misplaced.*

Reply: As pointed by the reviewer, we cite Ref 3 as “Particle hopping between optical potential wells has attracted much attention owing to its extensive involvement in many physical and biological processes, such as cell and DNA stretching [1–3], protein folding [4–7], chemical reactions [8, 9] and biomolecule sorting [10, 11].” in Line 47 Page 3.

In summary, we have addressed all comments of the reviewer. The manuscript and supplementary materials have been carefully corrected.

Reply to Reviewer 4

We are grateful to the Reviewer for the constructive comments and are delighted that the Reviewer is interested in our results and recommends the publication of this manuscript. We are happy to address all the comments.

Comment 1: *The paper propose a novel optofluidic chip for device based on nano-optofluidic lattice enables to manipulate individual bacterial cells in the flow stream. The approach represent an evolution optical chromatography and an enhancement respect to the previous paper, nor reported in the references, of the same Authors:*

Y. Z. Shi, et al. “High-resolution and multi-range particle separation by microscopic vibration in an optofluidic chip”, Lab Chip, 2017, 17, 2443-2450.

The use of multiple hotspots permits to add very powerful functionalities like particle bypassing, collision and aggregation. The experimental results on the binding affinity are very promising.

Reply: As suggested by the reviewer, in the revised manuscript, the Lab on a Chip paper is added as Ref [73] as “Particles and bacteria can be trapped in the microchannel based on the same principle as the optical chromatography [75–77].” in Line 118 Page 6.

Comment 2: *In the above paper the Authors use a micro-quadrangular lens in order to obtain a quasi-Bessel beam. In this paper the same micro-quadrangular lens is used in order to obtain the optical interference pattern with four hotspot. The optical interference pattern forms an array of non-uniform hotspots. The role of non-uniformity of the intensity between the four hotspots should be addressed in relationship to the three hopping mechanisms (i.e., particle bypassing, collision and aggregation).*

Reply: As suggested by the reviewer, the role of non-uniformity of the intensity between the four hotspots in relationship to the three hopping mechanisms are discussed in the Supplementary Information as “For hopping induced by particle bypassing, the pre-trapped particle blocks the upcoming particle and causes it to be trapped on the right side of the hotspot 1. Since the strength of the potential well in hotspot 2 or 4 is much stronger than that at the right edge of hotspot 1, the particle will eventually hop to hotspot 2 or 4 and the residence time of the particle in hotspot 1 is determined by the intensity of hotspot 2 or 4.”

For hopping induced by particle collision, the pre-trapped particle is directly pushed to hotspot 2 or 4 and the lateral distance (1.4 μm) between the two potential wells is the key factor inducing the hopping event. If the distance is too long, the pre-trapped particle may not be pushed to hotspot 2 or 4 from hotspot 1. On the contrary, if the distance is shorter, particle hopping will be easier.

For hopping induced by particle aggregation, both the lateral distance and the relative intensity of potential wells matter. The intensity of hotspot 2 or 4 will change the critical angle of the hopping since the particle has crossed the safe barrier as shown in Figure 4c.” in Page 17 of the Supplementary Information.

Comment 3: *The Authors shows results with particle and bacteria with size of about 1 micron. The particle the working size range of the proposed device should be analysed.*

Reply: As suggested by the reviewer, the working size range of the particle is analyzed and discussed as “The working size range of the particle is from 500 nm to 2 μm . The optical scattering force on the particle with 500 nm in diameter is about 3 pN, which is still much larger than the fluidic drag force (0.25 pN). However, a further decrease on the particle size will require a higher laser power (> 1 W). On the other hand, when the particle size is larger than 2 μm , the particle may occupy more than one hotspot because the lateral distance between two hotspots (e.g. hotspots 1 and 4) is only 1.4 μm . It will disturb the optical field significantly, and the particle hopping may not occur.” in Line 337 Page 15.

Comment 4: *Particle hopping triggered by lateral drag force in x-direction should be deeply discussed. The source of the lateral drag force should be explained (asymmetric hydrofocussing?)*

Reply: As pointed by the reviewer, in the revised manuscript, the lateral drag force is discussed as “The lateral force is generated when the velocity of upper sheath flow is slightly increased by approximately 20 $\mu\text{m/s}$, which causes the asymmetry of the hydrodynamic focusing. The lateral velocity should not be larger than 50 $\mu\text{m/s}$, or else, the particle will escape from the hotspot 4 without being captured by the hotspot 1. The drag force is estimated to be increased from 0.19 to 0.45 pN.” in the figure caption of Figure S2c.

Comment 5: *The reference list on optical chromatography should be improved.*

Reply: As pointed by the reviewer, the reference list of optical chromatography is added as “Particles and bacteria can be trapped in the microchannel based on the same principle as the optical chromatography [75–77].” in Line 118 Page 6.

In summary, we have addressed all comments of the reviewer. The manuscript and supplementary materials have been carefully corrected.

REVIEWERS' COMMENTS:

Reviewer #1 (Remarks to the Author):

I went through the revision and found it improved in line with reviewers' comments. The working principle was clarified further to facilitate readers' understanding, and the experimental data were presented with more and clearer labeling and explanation. I am happy with its current form.

Reviewer #3 (Remarks to the Author):

I appreciate the authors' responses to most of the issues that arose in review. One question that was brought up but remains is the role of anisotropic drag on the balance of forces. While optical forces certainly depend on shape [Ashkin, A., & Dziedzic, J. (1987) Optical trapping and manipulation of viruses and bacteria. *Science*, 235(4795), 1517–1520], the fluid drag on the bacteria varies significantly with flow orientation. With Brownian motion playing an important role at the length scales of concern here, could the authors comment on any sensitivity issues present with regard to trapping bacteria while simultaneously controlling particle hopping between adjacent potential wells?

Specific Comments

Please include error bars on the data reported in Tables 1 and S1.

Please fix Movie 4 and 5 (b) labels

Reviewer #4 (Remarks to the Author):

The paper is worth to be published

Manuscript ID: NCOMMS-17-21398A

Paper title: **Sculpting Nanoparticle Snooker for Single-bacteria-level Screening and Direct Binding-efficiency Measurement**

Authors: **Y. Z. Shi, S. Xiong, Y. Zhang, L. K. Chin, Y. -Y. Chen, J. B. Zhang, T. H. Zhang, W. Ser, A. Larson, L. S. Hoi, J. H. Wu, T. N. Chen, Z. C. Yang, Y. L. Hao, B. Liedberg, P. H. Yap, D. P. Tsai, C.-W. Qiu, and A. Q. Liu**

Reply to Reviewer 3

We are grateful to the Reviewer for the constructive comments and are delighted that the Reviewer is interested in our results and recommends the publication of this manuscript. We are happy to address all the comments.

Comment 1: *I appreciate the authors' responses to most of the issues that arose in review. One question that was brought up but remains is the role of anisotropic drag on the balance of forces. While optical forces certainly depend on shape [Ashkin, A., & Dziedzic, J. (1987) Optical trapping and manipulation of viruses and bacteria. Science, 235(4795), 1517–1520], the fluid drag on the bacteria varies significantly with flow orientation. With Brownian motion playing an important role at the length scales of concern here, could the authors comment on any sensitivity issues present with regard to trapping bacteria while simultaneously controlling particle hopping between adjacent potential wells?*

Reply: As pointed by the reviewer, in the revised manuscript, the description of bacteria manipulation from the shape and orientation in the optical and fluidic fields are added as "It is noted that the bacteria with rod shape should be aligned parallel to the flow direction (also the light propagating direction) in the laminar flow. Meanwhile, the bacteria with different shapes (diameter and length) experience different optical and fluidic forces [91, 92], which only causes the distributions of trapping positions of bacteria in Fig. 5 and Supplementary Fig. 11." in Line 338 Page 15.

Comment 2: *Please include error bars on the data reported in Tables 1 and S1.*

Reply: As pointed by the reviewer, the error bars on the data are added in **Table 1** and **Supplementary Table 1**.

Comment 3: *Please fix Movie 4 and 5 (b) labels*

Reply: As pointed by the reviewer, the labels "(a)" (should be "(b)") in Movie 4 and 5 are corrected.

In summary, we have addressed all comments of the reviewer. The manuscript and supplementary materials have been carefully corrected.